# Genome-Wide Identification and Expression Analysis of Growth-Regulating Factor Family in Sweet Potato and Its Two Relatives

**DOI:** 10.3390/genes15081064

**Published:** 2024-08-12

**Authors:** Wenhui Huang, Xiongjian Lin, Zhenqin Li, Jinglin Mai, Mengqin Hu, Hongbo Zhu

**Affiliations:** College of Coastal Agricultural Sciences, Guangdong Ocean University, Zhanjiang 524088, China; whhuang20@126.com (W.H.); l1843204672@outlook.com (X.L.); lizhenqiny@126.com (Z.L.); m2053968310@163.com (J.M.); hmm980725@163.com (M.H.)

**Keywords:** sweet potato, *Ipomoea trifida*, *Ipomoea triloba*, expression analysis

## Abstract

Growth-regulating factor (GRF) is a multi-gene family that plays an important role in plant growth and development and is widely present in plants. Currently, *GRF* gene members have been reported in many plants, but the *GRF* gene family has not been found in sweet potato. In this study, ten *GRF* genes were identified in sweet potato (*Ipomoea batatas*), twelve and twelve were identified in its two diploid relatives (*Ipomoea trifida*) and (*Ipomoea triloba*), which were unevenly distributed on nine different chromosomes. Subcellular localization analysis showed that *GRF* genes of sweet potato, *I. trifida*, and *I. triloba* were all located in the nucleus. The expression analysis showed that the expression of *IbGRFs* was diverse in different sweet potato parts, and most of the genes were upregulated and even had the highest expression in the vigorous growth buds. These findings provide molecular characterization of sweet potato and its two diploid relatives, the *GRF* families, further supporting functional characterization.

## 1. Introduction

*GRF* (growth-regulating factor) is an important transcription factor in plants. Its functions are involved in leaf, stem, flowering, seed, root development, growth control under stress conditions, and plant life regulation [1]. Although these genes are present in all eukaryotes and have a wide range of functions [2], the *GRF* genes are highly expressed in seedling tissues, such as root tips, flower buds, and young leaves, and relatively weakly expressed in mature tissues or organs [3]. Structural studies have shown that the *GRF* proteins consist of nine typical antiparallel α-helices [4]. They frequently assemble into either homodimers or heterodimers, with each GRF protein constituent of the dimer capable of engaging with a distinct protein [5]. This property enables them to unite two dissimilar proteins, forming a protein complex, which subsequently physically interacts with numerous protein clients integral to the biosynthesis and signaling pathways of key plant hormones. Moreover, mounting functional evidence underscores the pivotal regulatory roles played by GRF–target interactions [6]. Two conserved domains were found in the N-terminal of the *GRF* family proteins, and it was found that not all species of *GRF* genes contain both two conserved domains [7]. In this identification, except *ItbGRF5*, all *GRF* genes identified contained WRC (Trp, Arg, Cys, PFAM: PF08880) and QLQ (Gln, Leu, Gln, PFAM: PF08880) domains. The QLQ domain is considered to be a protein–protein interaction domain, which also interacts with *GRF* interacting factors (GIF), and the resulting complex acts as a transcriptional co-activator [8,9]. The WRC domain contains functional nuclear localization signals and DNA-binding motifs [10]. With the development of sequencing technology, an increasing number of plant genomes and transcriptomes are being revealed, leading to a broader scope of research in this area. The function of the first *GRF* member was determined by *OsGRF1* in rice, which is regulated by gibberellin, and overexpression of the *OsGRF1* transcription factor can promote the elongation of rice stems. Subsequently, the *GRF* gene family is being identified in species constantly, including 9 Arabidopsis [11], 12 rice [12], 30 wheat [3], 14 maize [10], 17 Chinese cabbage [13], 35 rapeseed [14], 10 Brachystis bispike [15], 25 pear [16], and so on.

In addition, several species of *GRF* gene families have been validated for their function. The overexpression of *AtGRF1* and *AtGRF2* led to the development of plants exhibiting larger leaves and cotyledons, accompanied by a delay in the bolting process of the inflorescence stem [11]. *AtGRF3* can promote the size of plant organs, do not cause morphological defects, that is, will not excessively increase the cost of plant defense, and also play a role in the early stage of panicle development [17]. *AtGRF5* can enhance the proliferation of leaf primordium cells, and the leaf growth is larger than that of wild type [18]. *AtGRF7* regulates abiotic stress by negatively regulating the expression of dehydration response element binding protein (DREB2A) [19]. *AtGRF9* has the function of reducing the leaf size by restricting the number of cells in the leaf primordiums, while the size of the leaf cells remains unchanged [20]. *OsGRF1* plays a regulatory role in stem elongation induced by gibberellin, also mainly at the early stages of panicle development [8,21]. The mutation of *OsGRF4* can generate larger grain size and enhance higher grain yield in rice [22]. *OsGRF6* can adjust the development of auxiliary branches and spikelets and greatly improve the grain yield indirectly [23]. *OsGRF7* has been shown involved in rice tiller determination to regulate the biosynthesis of strigolactone [24]. The overexpression of *OsGRF8* can resist the infection of brown planthopper (BPH) on rice and increase rice yield; it can also be involved in regulating rice particle size, improving crops, and increasing yields [25,26]. As with *OsGRF6*, *OsGRF8* was able to express largely in developing panicles, and mainly at the early stages of panicle development, while *OsGRF7* and *OsGRF9* mainly at the later stages [21]. *OsGRF11* can also play a role in the early panicle development and participate in reproductive function [21]. Overexpression of *BrGRF8* significantly increases the shoot and root fresh weight, seedling root length, and lateral root number and significantly reduce the nitrate content under nitrate-poor and nitrate-rich conditions in Arabidopsis, which is related to its extensive regulation with N uptake, utilization, and signaling [27]. Silencing *CsGRF04* can significantly reduce the resistance to salt stress and cold stress but can improve the drought tolerance in Citrus [28]. *LsaGRF5* may play a role as a transcription factor in the nucleus through subcellular localization observation and transactivation assays, and the overexpression of *LsaGRF5* can stimulate leaf growth and lead to leaf enlargement [29]. The PbGRF18 has the ability to enhance the sugar content found in both the leaves and fruits of tomatoes [16]. *MdGRF6* enhances the sensitivity of apples to salt stress by modulating the activity of antioxidant enzymes and regulating the expression of genes involved in the salt stress response [30].

Sweet potato [*Ipomoea batatas* (L.) Lam.] is an annual herb belonging to the Convolvulaceae family, which is the seventh most important food crop in the world, also an important energy crop for its high edible, feed, and medicinal values [31]. The hexaploid nature of sweet potato (2n = 6x = 90), characterized by a high degree of heterozygosity and general self-incompatibility, leading to outcrossing polyploidy, poses numerous obstacles to conventional breeding methods [32]. So, its productivity and quality are often limited by abiotic and biological stresses, which affect greater economic losses. The use of genetic engineering to change molecular breeding mechanisms has been shown to have great potential to improve the resistance of sweet potatoes to these pressures and improve quality and yield [33]. In this study, the *GRF* gene family of sweet potato was identified by bioinformatics, and its physicochemical property, collinearity, chromosome distribution, phylogenetic relationship, conserved motifs, and cis-regulatory elements of promoters were analyzed. Furthermore, the candidate *GRF* genes were quantitatively analyzed and their expression patterns were assessed. The results provided a theoretical basis for understanding the function and molecular breeding of the *GRF* gene in sweet potato.

## 2. Results

### 2.1. Identification of GRFs in Sweet Potato and Its Two Diploid Wild Relatives

Ten, twelve, and twelve *GRF* genes were identified in *I. batatas*, *I. trifida*, and *I. triloba*, which were named *IbGRF1-10*, *ItfGRF1-12*, and *ItbGRF1-12*, respectively, according to their positions on the chromosome. In sweet potato, CDS length ranged from 1071 bp (*IbGRF10*) to 3672 bp (*IbGRF7*), genome length ranged from 2444 bp (*IbGRF5*) to 7869 bp (*IbGRF8*), and the protein sequence ranged from 356 aa (IbGRF10) to 611 aa (IbGRF7). The molecular weight (MW) was 39.16 kDa to 64.87 kDa, and the isoelectric point (pI) was 5.89 (IbGRF3) to 9.22 (IbGRF5). All IbGRFs were unstable with an instability coefficient greater than 45. They were hydrophilic proteins with negative hydrophilic index, and the best hydrophilic protein was IbGRF5 (−0.762). Subcellular localization predictions showed that all identified sweet potato *IbGRFs* were located in the nucleus (Table 1).

In *I. trifida*, the CDS length ranged from 957 bp (*ItfGRF6*) to 1815 bp (*ItfGRF7*), the genome length ranged from 1979 bp (*ItfGRF10*) to 4507 bp (*ItfGRF5*), and the protein sequence ranged from 318 aa (ItfGRF6) to 604 aa (ItfGRF7). The MW was 35.42 kDa (ItfGRF6) to 64.02 (ItfGRF7) kDa, and the pI was 5.89 (ItfGRF12) to 9.1 (ItfGRF10). In *I. triloba*, the CDS length ranged from 957 bp (*ItbGRF6*) to 1830 bp (*ItbGRF7*), the genome length ranged from 2075 bp (*ItbGRF12*) to 4623 bp (*ItbGRF5*), and the protein sequence was 319 aa (*ItbGRF6*) to 609 aa (*ItbGRF7*). With a MW of 35.24 kDa (ItbGRF5) to 64.63 kDa (ItbGRF7), the pI ranged from 6.26 (ItbGRF12) to 9.22 (ItbGRF10). All IbGRFs were unstable with an instability coefficient greater than 45. It can be seen that both ItfGRFs and ItbGRFs were soluble in water, and the hydrophilic indexes were negative. The best hydrophilic indexes were ItfGRF4 and ItbGRF4, respectively. Subcellular localization prediction revealed that all identified *ItfGRFs* and *ItbGRFs*, like *IbGRFs*, were located in the nucleus (Table 1).

Chromosome localization results showed that *GRF* genes from *I. batata*, *I. trifida*, and *I. triloba* were unevenly distributed on 9 chromosomes, respectively. Some chromosomes contained two *GRF* genes, such as LG5 of *I. batata*, Chr03, Chr08, and Chr12 of *I. trfida*, and Chr03, Chr08, and Chr12 of *I. triloba* (Figure 1 and Appendix A).

### 2.2. Collinearity Analysis of GRF Genes and Ka/Ks Analysis

Gene replication, including tandem replication, fragment replication, and whole genome replication, is the primary driving force of plant evolution [34]. In order to reveal the replication mechanism of *IbGRFs*, *ItfGRFs*, and *ItbGRFs* gene families, we analyzed *GRFs* gene pairs and performed intraspecific collinearity analysis using TBtools method (v.2.084). It showed that *I. batatas*, *I. trifida*, and *I. triloba* generate 6 pairs, 8 pairs, and 9 pairs (Figure 2a–c). It showed that there was more collinearity between species, greater relationships, and greater homology, and these genes originated from ancestral genes (Figure 2d).

Furthermore, we calculated the Ka/Ks ratio to estimate the selection pressure and gene evolution rate among duplicate *GRF* genes [35]. The results showed that the Ka/Ka ratio of *GRFs* duplicate genes ranged from 0.136 to 0.404, with an average value of 0.221. The Ka/Ks ratio of all gene pairs was less than 1, which meant that these genes duplicated and evolved in purification selection. Also, according to the approximate dates of *GRFs* calculation formula of the obtained gene duplication events, the duplication events of *GRFs* occurred in about 128.374 Mya to 667.871 Mya, with an average of 329.591 Mya (Appendix A).

### 2.3. Phylogenetic Relationship Analysis of GRFs in Sweet Potato and Its Two Diploid Relatives

Based on the identified *GRFs* that were downloaded from NCBI, MEGA7.0 software was used to construct a phylogenetic tree and analyze the evolutionary relationships of GRFs among 10 *I. batatas*, 12 *I. trifida*, 12 *I. triloba*, 9 *Arabidopsis thaliana*, 10 *Oryza sativa*, and 17 *Brassica rapa*. The results showed that similar sequences with evolutionary relationships were clustered together. According to the relationship between evolution and development, 70 GRFs were divided into 4 groups (Figure 3). In general, all *IbGRFs*, *ItfGRFs*, and *ItbGRFs* were clustered together with their corresponding homologues (Figure 3).

### 2.4. Conserved Motifs, Conserved Domain and Exon–Intron Structure Analysis of GRFs from I. batatas and Its Two Relatives I. trifida and I. triloba

The MEME online website was used to predict the conserved motifs of *GRFs* and a total of 10 conserved motifs were identified and the sequence logos were obtained (Figure 4). Also, a hitdate file was downloaded from the NCBI database to predict conserved domains (Figure 5c). As shown in Figure 5, the 34 GRFs were divided into seven groups. There were certain differences in the number and distribution of motifs in each group of GRFs; each gene contained 4 to 10 motifs, and all members had motif 1. Group 1 of GRFs lacked motif 7 and motif 9, Group 2 had ten motifs, Group 3 lacked motif 6 and motif 9, Group 4 lacked motif 9, Groups 5, 6, 7 and 8 had a small number of motifs, Group 9 lacked motif 5, motif 7, motif 10, where ItbGRF5 in Group 9 also lacked motif 2, Group 10 lacked motif 5, motif 6, motif 7, motif 10, Group 11 lacked motif 5, motif 6, motif 7, motif 9, motif 10, and Group 12 lacked motif 6, motif 7, motif 9, motif 10 (Figure 5b). Therefore, the *GRF* genes exhibited variations during their evolutionary processes.

Furthermore, we analyzed the structure of *GRFs* by TBtools software (v.2.084). According to the results, we can find a clear difference in the number of exons and introns in *GRFs*. The number of introns in the *GRF* genes ranged from 2 to 7, while the number of exons ranged from 3 to 8. Most of these genes had three introns (27 out of 34), followed by four introns (5 out of 34), and most of these genes had four exons (17 out of 34), followed by three exons (9 out of 34) (Figure 5d). These results suggested that the similarities and differences of genes during evolution may be related to intron–exon differences.

### 2.5. Analysis of Putative Cis-Regulatory Elements of GRFs Promoters from I. batata and Its Two Diploid Relatives I. trifida and I. triloba

Cis-acting elements are transcription factor DNA binding sites and other regulatory motifs with special functions in the same DNA molecule, which hold a crucial position in the regulation of gene transcription initiation. Therefore, we extracted the 2000 bp promoter sequence upstream of the *GRF* genes start codon from TBtools software (v.2.084) and used the PlantCARE online tool to predict the regulatory characteristics of expression in *GRF* genes. As shown in Figure 6, we found a number of endogenous hormone response elements, including abscisic acid responsiveness (ABRE), MeJA responsiveness (CGTCA-motif and TGACG-motif), gibberellin responsiveness (GARE-motif, P-box, and TATC-box), salicylic acid responsiveness (TCA-element), and auxin responsiveness (AuxRR-core and TGA-element), that indicated *GRFs* were involved in plant hormone expression. In addition, several abiotic stress-responsive elements, such as anaerobic induction elements (ARE and GC-motif), low-temperature responsiveness element (LIR), drought-inducibility element (MBS), and defense and stress responsiveness element (TC-rich repeats), were found in promoters. Therefore, we also found that some growth and development regulatory and biosynthesis elements, covering meristem expression (CAT-box), seed-specific regulation (RY-element), cell cycle regulation (MSA-like), zein metabolism (O2-site), circadian control (circadian), endosperm expression (GCN4_motif), and palisade mesophyll cells differentiation (HD-Zip 1) in *GRFs*. The results showed that *GRF* genes were closely related to the growth and development of sweet potato and biological processes such as biological and abiotic stress. In addition, it also found that protein binding site, regulatory element, wound responsiveness, and elicitor-mediated activation. Interestingly, a total of 339 light responsiveness elements were identified, with the most prevalent elements among the three species. This suggested that *GRF* genes were involved in light regulation (Figure 6).

### 2.6. Expression Analysis of GRFs in I. batata, I. trifida and I. triloba

#### 2.6.1. Expression Analysis under Hormone Stress

To investigate the potential biological function of *GRF* genes in plant hormone signal processing, we utilized the “Xushu 18” RNA-seq data obtained from the NCBI database (PRJNA511028) and characterized the expression profiles of 10 *IbGRF* genes across three distinct tissues following treatment with ABA, SA, and MeJA. In fibrous roots, *IbGRF3* and *IbGRF10* were upregulated after the ABA treatment, *IbGRF2*, *IbGRF4*, and *IbGRF7*-9 were upregulated after the SA treatment, and *IbGRF2*-*3*, *IbGRF6*-10 were upregulated after the MeJA treatment. In stems, the transcription of *GRF* genes was different under diverse hormone stress. For instance, *IbGRF1* was downregulated, while *IbGRF2*, *IbGRF4*, and *IbGRF8*-*10* were upregulated after the three hormone treatments. In leaf, all *IbGRFs* were upregulated after SA and MeJA treatments, but after the ABA treatment, except for *IbGRF3* and *IbGRF7*, the left genes were upregulated or had no significant change (Figure 7).

Furthermore, RNA-seq data of *I. trifida* and *I. triloba* were used to analyze the expression patterns of *ItfGRFs* and *ItbGRFs* under ABA, GA3, and IAA treatments. After the ABA treatment, only *ItfGRF2* and *ItfGRF5* were downregulated, and the others were upregulated or did not show significant changes. Under the GA3 treatment, only *ItfGRF4* and *ItfGRF11* were upregulated, while the other was downregulated. Under the IAA treatment, eight *ItfGRFs* were downregulated, and *ItfGRF2*, *ItfGRF6*, *ItfGRF10*, and *ItfGRF11* were upregulated. In *I. triloba*, *ItbGRF3*, *ItbGRF4*, *ItbGRF8*, and *ItbGRF9* were upregulated, and the others were downregulated or did not show significant changes after ABA treatment. Under the GA3 treatment, *ItbGRF4*, *ItbGRF8*, *ItbGRF9*, and *ItbGRF11* were upregulated, except *ItbGRF3* and *ItbGRF10*, and the others were downregulated. Under the IAA treatment, only *ItbGRF7*, *ItbGRF10*, and *ItbGRF12* were upregulated, and the others were upregulated or had no significant change (Appendix A).

#### 2.6.2. Expression Analysis under Cold Stress

The expression profiles of 10 *IbGRFs* were detected by transcriptome data of cold−tolerant “Liaohanshu 21” and cold−sensitive “Shenshu 28”, which was studied in a previous report after cold stress [36]. As shown in Figure 8, except for *IbGRF9*, only *IbGRF6* were upregulated in “Shenshu 28” after cold stress, while the rest of the genes were all downregulated. And *IbGRF1*, *IbGRF5*, *IbGRF6*, and *IbGRF10* were upregulated in “Liaohanshu 21” after cold stress, while *IbGRF2*, *IbGRF3*, *IbGRF4*, *IbGRF7*, and *IbGRF8* were upregulated (Figure 8). 

Additionally, the expression patterns of *ItfGRFs* and *ItbGRFs* were analyzed also. *ItfGRF1*, *ItfGRF4*, *ItfGRF5*, *ItfGRF7*, *ItfGRF10*, and *ItfGRF11* were upregulated after cold stress, whereas half−leaf *ItfGRFs* were downregulated. After a cold treatment, only *ItbGRF11* was upregulated; the remaining genes were all downregulated (Appendix A).

#### 2.6.3. Expression Analysis under Heat Stress

The expression profiles of 10 *IbGRFs* were detected by transcriptome data of heat−tolerant “Guangshu 87” and heat−sensitive “Ziluolan”, which were studied in a previous report after heat stress. As shown in the figure, the transcripts of fibrous root in the “Ziluolan” cultivar were downregulated except for *IbGRF7*, but in the “Guangshu 87” cultivar, only *IbGRF9* was upregulated, while the other *IbGRFs* were downregulated. The transcripts of *IbGRF1*, *IbGRF3*, *IbGRF9*, and *IbGRF10* in tuberous root were upregulated in the “Ziluolan” cultivar, while the other *IbGRFs* were downregulated. Otherwise, in the “Guangshu 87” cultivar, only *IbGRF3*, *IbGRF6*, and *IbGRF10* were upregulated, while the other *IbGRFs* were downregulated (Figure 9).

Furthermore, the expression patterns of both *ItfGRFs* and *ItbGRFs* were additionally analyzed under conditions of heat stress. After heat treatment, only *ItfGRF5* and *ItfGRF7* were upregulated. Five *ItbGRFs* were upregulated, while *ItbGRF1*, *ItbGRF2*, *ItbGRF6*, *ItbGRF7*, *ItbGRF9*, *ItbGRF10*, and *ItbGRF11* were downregulated (Appendix A).

#### 2.6.4. Expression Analysis under Salt and Drought Stresses

To explore the possible roles of *IbGRFs* in an abiotic stress response, we analyzed the expression patterns of *IbGRFs* under salt and drought treatments. As shown in the Figure 10, in flower organs, the expression of almost all *IbGRFs* genes was downregulated after salt and drought stresses, except *IbGRF2*, *IbGRF3*, *IbGRF6*, and *IbGRF10*, which were upregulated after drought stress. In firewood root, *IbGRF2* and *IbGRF8* were both upregulated after salt and drought stresses. In fruit, more than half of the *IbGRFs* were upregulated after salt treatment, while more than half of the *IbGRFs* were downregulated after drought treatment. In leaf, only *IbGRF4*, *IbGRF6*, and *IbGRF10* were upregulated, while the others were downregulated after salt and drought stresses, except *IbGRF7* and *IbGRF9*, which were upregulated after salt treatment but downregulated after drought treatment. In fibrous root, five *IbGRFs* were upregulated while two *IbGRFs* were downregulated after two kinds of stresses. In primary root, only *IbGRF6* was upregulated after salt stress, and *IbGRF5*, *IbGRF6*, and *IbGRF7* were upregulated after drought stress. In stem, all *IbGRFs* were downregulated after salt stress but not after drought stress. In root tuber, only *IbGRF4* and *IbGRF6* were upregulated after salt stress, and only *IbGRF3* and *IbGRF8* were upregulated after drought stress (Figure 10). 

To sum up, the expression of *IbGRFs* in most parts was downregulated under the salt stress. However, the expression of *IbGRFs* in fruit was upregulated mostly, which indicated that there are other factors in fruit that can promote the resistance of *GRFs* to salt stress and promote their expression. But the rule was not obvious under drought stress.

In addition, so as to explore the function of *I. trifida* and *I. triloba*, we examined the expression patterns of them under salt and drought stresses [37]. In *I. trifida*, *ItfGRF4* and *ItfGRF9* were upregulated under both stresses, while *ItfGRF1*, *ItfGRF5*, *ItfGRF7* and *ItfGRF12* were downregulated. In *I. triloba*, *ItbGRF4* and *ItbGRF9* were upregulated under both stresses, while *ItbGRF1*, *ItbGRF3*, *ItbGRF7*, *ItbGRF10*, and *ItbGRF12* were downregulated (Appendix A).

#### 2.6.5. Expression Analysis in Various Tissues

The transcript levels of *IbGRFs* identified by qPCR were detected in nine organs: flower, flower bud, fibrous root, firewood root, primary root, tender stem, old stem, tender leaf, and old leaf. The results revealed that the expression of *IbGRF4* was higher in flowers and buds than in other tissues, which had a tissue-specific expression pattern. And the expression of *IbGRFs* was highest in the flower bud (Figure 11). According to the relative expression levels of *IbGRFs* in 9 tissues, old leaf was used as a control group; the expression of six *IbGRFs* (i.e., *IbGRF1*, −3, −4, −6, −9, −10) was upregulated, and the amount of expression was the highest. It was also found that *IbGRFs* were all have certain expression in flower and bud (Figure 12). These results indicated that *IbGRFs* are conducive to plant growth and development and express in large quantities in the meristem. The qPCR primers are detailed in Appendix A.

### 2.7. Protein Interactions Network of GRFs in Sweet Potato

A sweet potato GRF protein interactions network was constructed based on the *Arabidopsis* protein interactions model. Protein interaction maps showed that all ten IbGRFs were homologous with AtGRFs, and all IbGRFs interacted with each other. For instance, IbGRF1 and IbGRF10 were homologous with GRF5-2, IbGRF2 and IbGRF4 were homologous with GRF2-2, IbGRF3 was homologous with GRF7-2, IbGRF5 were homologous with GRF9, IbGRF6 was homologous with GRF8-2, IbGRF7 and IbGRF8 were homologous with GRF1, and IbGRF9 was homologous with GRF3-2. Additionally, these IbGRFs also interacted with other functional proteins, such as transmembrane protein F4HT79 ARATH, domain-containing protein MJB21.7 and MVI11, and GIFs, which were GRF-interacting factors and transcription coactivators (Figure 13). In conclusion, IbGRFs may also be involved in the transmembrane transport of substances and participate in the composition of the domain. What is more, the figure confirmed that GRFs had a non-negligible relationship with GIF, which are bona fide partner proteins that together form a unique transcriptional complex specific to plants. Also, the GRF–GIF duo imparts a meristematic specification state to the primordial cells of both vegetative and reproductive organs, ensuring a steady supply of cells for organogenesis and successful reproduction [38].

### 2.8. Transcript Factors Network of Sweet Potato GRF Genes

The results of TF analysis in *IbGRFs* showed that a total of 578 TFs were identified and distributed in 45 TF families, and Figure 14 showed only 40 TFs of them, distributed in 11 families (ABI3, AP2, BHLH, C4-GATA-related, ERF/DREB, IDD, Jumonji, MIKC, RAV, SBP, MYB-related). Among them, the ERF/DREB TF family had 79 members, which was the most enriched; also, the number of members varies. What is more, the number of 10 *IbGRFs* genes targeted by TF was equal, and the members were similar, indicating that the functions of these ten genes were similar and functional redundancy existed in sweet potato (Figure 14).

## 3. Discussion

### 3.1. Identification and Evolution of Sweet Potato and Its Two Diploid Relatives I. trifida and I. triloba

The development and maturation of sequencing technology have allowed many plant genomes to be analyzed, thereby facilitating the identification and analysis of plant gene families at the whole genome level. Owing to the complexity of the hexaploid sweet potato genome, it is frequently analyzed in conjunction with its two diploid relatives, *I. trifida* and *I. triloba*, which is also beneficial to analyze the unity and specificity of structure and function and evolutionary relationships [39,40,41]. In our study, 10, 12, and 12 *GRF* genes were identified in *I. batatas*, *I. trifida*, and *I. triloba*, respectively. In general, the number of *GRF* transcription factor genes in terrestrial plants ranges from 8 to 20, but the number is lower in bryophytes, algae, and other plants [7]. Some studies suggest that the significant evolutionary expansion of *GRF* gene family members was due to enhanced adaptation to complex environments [42]. These genes were scattered across nine chromosomes, some of which contain two *GRF* genes, but were presented as single genes rather than clusters, indicating that they each had different biological regulatory and stress response functions.

According to previous studies, amplification of the *GRF* family occurred primarily through gene replication, including whole genome duplication (WGD) or tandem duplication [43]. The occurrences of fragment duplication were disseminated across the respective chromosomes, and these duplicated gene fragments were able to persist in diverse species via WGD, which was consistent with the previous research findings [44]. What is more, demonstrated the amplification of the *GRF* gene family mainly depended on fragment replication. This type of replication enriched the diversity of gene families and accelerated species evolution, improved the adaptability of evolved species to the environment and its own survival ability, promoted the morphological evolution of species, and played an important role in the expansion of *GRF* gene families (Figure 2).

Differences in gene structure and function are associated with exon/intron gain/loss events [45]. In our study, the number of introns in the *GRF* genes ranged from 2 to 7, while the number of exons ranged from 3 to 8. It seemed that the number of exons/introns in *GRF* genes was not conserved, and the differences might be caused by chromosome rearrangement and fusion, which in turn led to different biological functions of *GRF* genes [43]. However, in the same group, the number was conserved, indicating that the gene structure and function were similar in the same group (Figure 5d).

Cis-acting elements act as binding sites for various regulatory factors and proteins, thereby affecting gene expression. In the *GRF* genes, there were a large number of light response elements and hormone stress response elements, indicating that they were involved in light response and hormone signaling. In addition, various abiotic stress elements can also be found in *GRF* genes (Figure 6), suggesting that the regulatory role and expression of the *GRF* gene might be affected by environmental factors [46].

### 3.2. GRFs Are Involved in Plants Growth and Development Hormonal Regulation

Studies have shown that *GRF* genes play an important role in the regulation of plant growth and development, stress response, and other biological processes [8,47]. Just like previous studies in *Arabidopsis*, rice, corn, and even other plants, most of the *AtGRFs*, *OsGRFs*, and *ZmGRFs* are strongly expressed in actively growing and developing tissues, such as shoot tips, flower buds, and roots, but weakly in mature stem and leaf tissues [10,11,12]. In this study, quantitative analysis of candidate *GRF* genes also showed that the expression of *GRF* genes in young tissues or organs (flower bud) of sweet potato was higher than that in mature tissues or organs. As a result, genes of the *GRF* family mainly expressed in certain organs or tissues might play important roles in the growth and development of these organs or tissues.

GA was involved in various physiological activities and hormone regulation of plants [48]. Previous studies have shown that GA3 treatment induces upregulation of OsGRF1/2/3/7/10/12 and represses the expression of *OsGRF9* in rice but results in reduced expression of most *GRFs* in cabbage [12,13]. In this study, we also found some gibberellin responsiveness cis-element, which implicated that gibberellin will induce or inhibit the function of *GRFs* in the plant. Furthermore, in one previous study, the KNOX protein negatively regulated gibberellin to increase its level in immature tissues and organs and then participated in the establishment and maintenance of plant meristems [47,48]. In the other study, *GRF* proteins acted as repressors and downregulators of KNOX gene expression [49]. As previously reported in rice and Chinese cabbage [12,13], we conclude that most *IbGRFs* transcription is induced by GA3 treatment. These results suggested that the *GRF* genes may function in maintaining or promoting cell proliferation in plants by a feedback regulation mechanism, in which the *GRF* genes positively regulate the production of GAs, and GAs in turn upregulate *GRF* gene expression. Surprisingly, *GRFs* are able to interact with a negative regulator of GA signaling, DELLA. What’s more, *GRF* activity was balanced by an antagonistic regulatory relationship with DELLA growth inhibitors [50]. The conclusion was contrary to most studies showing that *GRF* was a positive factor in promoting growth.

## 4. Materials and Methods

### 4.1. Identification and Physicochemical Properties of GRF Family Members

The protein sequences and annotation files of sweet potato, *I. trifida*, and *I. triloba* were obtained from the Sweet Potato Genomics Resource (http://sweetpotato.plantbiology.msu.edu/, accessed on 21 February 2024) and the Ipomoea Genome Hub (https://sweetpotao.com/, accessed on 21 February 2024). The *Arabidopsis* GRF proteins were downloaded from the TAIR database (https://www.arabidopsis.org/, accessed on 2 Match 2024), the rice. GRF proteins were downloaded from NCBI (https://www.ncbi.nlm.nih.gov/, accessed on 8 Match 2024), and the *Brassica* GRF proteins were downloaded from the BRAD database (http://brassicadb.cn/, accessed on 9 Match 2024). In order to identify the IbGRFs, we adopted the following methods. First, we developed an experimental method and fully consulted the literature to obtain the GRF protein sequence of the model plant *Arabidopsis*. Second, HMMsearch from NCBI was used to conserve the domain of AtGRFs (Pfam accession numbers: PF08879 and PF08880) [51]. Then, *Arabidopsis* GRF protein sequences were searched in the sweet potato variety Taizhong protein sequence library using the Blastp function in Bioedit software (v.7.1.3.0) with an E-value threshold of 1 × 10^−10^ [52]. After searching, there were 835 genes with similar parts of *Arabidopsis GRF* gene structure. Afterwards, the domain of all the IbGRFs was identified by the CD-Search tool in the NCBI database and the Simple Modular Architecture Research Tool (SMART) (https://smart.embl.de/, accessed on 22 April 2024) [53]. Genes that do not contain conserved domains were deleted, while genes with intact domains were retained. Finally, 10 *GRF* gene family candidate genes of sweet potato were obtained, and the ratio of gene number to blastp result was 0.20. The Protein Parameter Calc function of TBtools software (v.2.084) was used to calculate protein physicochemical parameters [54]. Wolf PSORT (https://wolfpsort.hgc.jp/, accessed on 24 April 2024) was used to predict subcellular localization [55].

### 4.2. Collinearity Analysis of GRF Genes and Calculation of Ka/Ks Value

The MCScanX function of TBtools software (v.2.084) was used to construct inter-species and intra-species collinearity relationships, and Synteny Plot was used to visualize the relationships [56]. Similarly, we used TBtools 2.084 with default parameters to calculate the Ka/Ks ratio to estimate the rate of evolution of *GRF* genes over repetition. (Ka: nonsynonymous substitution rate, Ks: synonymous substitution rate). The estimated separation time (T, Mya: million years ago) was calculated according to the previous report [57].

### 4.3. Phylogenetic Analysis of GRFs

The phylogenetic analysis tree of the GRFs of *Arabidopsis*, rice, and Chinese cabbage, which were acquired, and *I. batatas*, *I. trifida*, and *I. triloba*, which were screened, was constructed using the ClustalW function in MEGA7.0 software with the default parameter neighbor-joining (NJ) method, and bootstrapping was performed with 1000 replicates [58]. After obtaining the phylogenetic tree, the online pretty print tool Evolview (https://www.evolgenius.info/evolview/, accessed on 28 April 2024) and Adobe Illustrator 2022 (AI) were used to beautify the phylogenetic tree [59].

### 4.4. Conserved Motifs and Gene Structure Analysis of GRFs

The conserved motif of GRF protein sequence was analyzed by the MEME (v.5.5.5) online tool (https://meme-suite.org/meme/tools/meme, accessed on 17 March 2024), and the number of motifs was set to 10, while default settings were adopted for other parameters [60]. Furthermore, the hitdata file of protein sequence and the Newick file of phylogenetic tree were downloaded from the NCBI online website and MEGA 7.0. Finally, the *GRF* genes phylogenetic tree, gene structure, and conserved motifs were visualized on the Gene Structure View function of TBtools software (v.2.084) to obtain the conserved motif and gene structure map [54].

### 4.5. Analysis of Putative Cis-Regulatory Elements of GRFs

The upstream promoter 2000 bp was extracted by TBtools software (v.2.084) and then submitted to the PlantCare (https://bioinformatics.psb.ugent.be/webtools/plantcare/html/, accessed on 25 April 2024) online website, analyzing the cis-components of *GRFs* [61]. Then, processed file data were obtained through personal email, and the Simple BioSequence Viewer of TBtools software (v.2.084) was used to visualize the cis-regulatory element figure [54].

### 4.6. Protein Interactions Analysis of Sweet Potato GRFs

Initial parameters were set on the online website STRING (https://cn.string-db.org/, accessed on 23 April 2024) to execute the potential protein interactions between model plant *Arabidopsis Thaliana* and sweet potato GRF proteins [62]. After obtaining the initial interaction figure, Cytoscape software (v.3.9.1) was used to beautify the protein–protein interaction network [63].

### 4.7. Transcript Factors Regulatory Network Analysis of Sweet Potato GRFs

Based on the homologous relationship between model plant Arabidopsis thaliana and sweet potato, the JASPAR online website (https://jaspar.elixir.no/, accessed on 25 April 2024) was used to predict the TFs of sweet potato *GRF* genes [64]. The Cytoscape software version 3.9.1 was used to visualized the potential TF regulatory networks [63].

### 4.8. Transcriptome Analysis

Five transcriptome bio-project datasets were chosen for the sweet potato *GRF* gene expression profile analysis. Of which, two bio-project datasets (PRJNA511028 for hormone and PRJNA987163 for cold) were downloaded from the NCBI database. Another three were our in-house (unpublished) sweet potato heat treatment, salt treatment, and drought treatment. Among them, “Xushu 18” was for hormonal treatment, cold-tolerant “Liaohanshu 21” and cold-sensitive “Shenshu 28” for cold treatment, heat tolerant “Guangshu 87” and heat-sensitive “Ziluolan” for heat treatment, and salt-tolerant and drought-tolerant “Guangshu 87” for salt and drought treatment. Additionally, the gene expression data of *I. trifida* and *I. triloba* were downloaded from the Sweet Potato Genomics Resource (http://sweetpotato.plantbiology.msu.edu/, accessed on 15 March 2024). The *GRF* expression was measured in fragments per kilo base of exon per million fragments mapped (FPKM) [65]. The heat maps of expression were constructed by TBtools software (v.2.084) [54].

### 4.9. Quantitative Analysis of Candidate IbGRF Genes

The sweet potato (*I. batatas*) cultivar “Jishu 26” was used for qRT-PCR analysis in this study. Sweet potato plants were cultivated at the experimental field of Guangdong Ocean University, Guangdong, China. For tissue expression, the flower, flower bud, tender leaf, old leaf, tender stem, old stem, primary root, firewood root, and tuberous root tissues were sampled from 3-month-old “Jishu 26” planted in the field. Among them, tender and old leaves are the first spreading leaf and the fifth leaf from the apical meristem; the tender and old stem are the first and tenth segments from the meristem. After the tissues of different parts of sweet potato were taken from the experimental field, they were covered with dry ice after being quickly frozen with liquid nitrogen. Then extracted RNA by the TRIzol method (Invitrogen, Carlsbad, CA, USA) in the laboratory, which was then transcribed into cDNA [66]. Primers were designed using Prime-BLAST of the NCBI database and sent to the company to synthesize primers. The qRT-PCR reaction was performed utilizing the Bio-Rad system, adhering to the specified thermal cycling protocol: Initiating with a 3 min pre-degeneration at 95 °C, subsequently proceeding through 40 cycles consisting of a 10 s denaturation phase at 95 °C and a 30 s annealing phase at 60 °C. The reaction concluded with a 5 s final extension step at 65 °C, followed by a gradual cooling to 95 °C at a rate of 0.5 °C per step [67]. Each sample was replicated three times, adhering to Dingfa’s methodology, which employed the *IbARF* gene as an internal reference. For quantification, we utilized the 2^−ΔΔCT^ method to determine the relative transcript levels [68].

## 5. Conclusions

In this study, 34 *GRF* genes were identified in the sweet potato and its two diploid relatives. By analyzing the development and evolution of *GRF* genes, we found that *IbGRFs* experienced frequent duplication leading to complex functions. Analysis of conserved motifs and gene structure showed that *GRFs* had conserved and dispersed characteristics in the evolutionary process. The results of the study indicated that the *GRF* genes were involved in light-responsive expression, which was consistent with their role as growth-regulating factors involved in plant photosynthesis and growth and development. Expression analysis revealed the expression patterns of *IbGRFs* in different sweet potato parts, reflecting the expression diversity and thus reflecting the functional diversity and regulatory role of *IbGRFs*. In Figure 15, we clearly demonstrated *GRF* genes worthy of further study in plant growth and development in the form of conductive graph, including *IbGRF1* and *IbGRF4*, and the expression patterns of *GRF* gene under abiotic and hormonal stress. This study provides a solid foundation for further exploring the molecular evolutionary mechanism and potential biological functions of the sweet potato *GRF* gene family.

## Figures and Tables

**Figure 1 genes-15-01064-f001:**
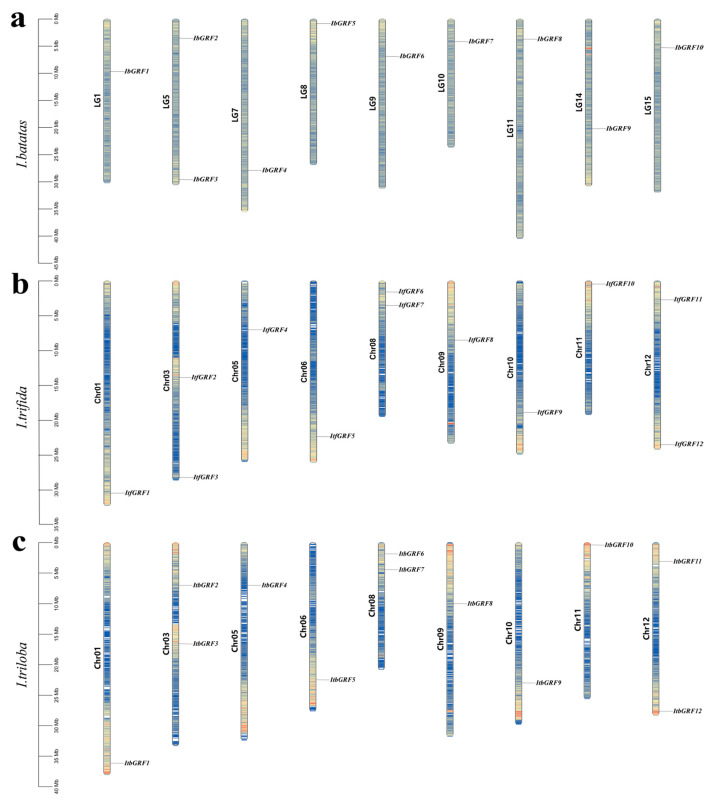
Chromosomal localization and distribution of *I. batatas* (**a**), *I. trifida* (**b**) and *I. triloba* (**c**). The bars represent chromosomes, with chromosome numbers on the left and gene names on the right. The number line on the left gives a visual view of chromosome length, and the number line on the left indicates species. Blue, yellow, and red represent the size of the gene density on the chromosome, from small to large, respectively.

**Figure 2 genes-15-01064-f002:**
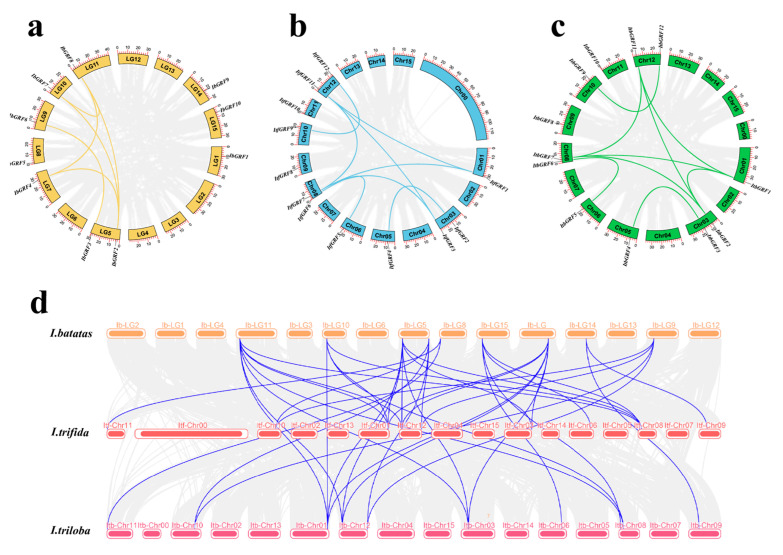
Localization and collinearity analysis of the *GRF* genes in *I. batatas* (**a**), *I. trifida* (**b**), and *I. triloba* (**c**), with yellow, blue, and green blocks representing chromosomes, and colored lines connecting two duplicate genes. (**d**) Collinearity analysis between the three species, the color block represents the chromosome, and the gene pairs with repeated relationships are connected by a bluish violet line, representing the common relationship between the three species.

**Figure 3 genes-15-01064-f003:**
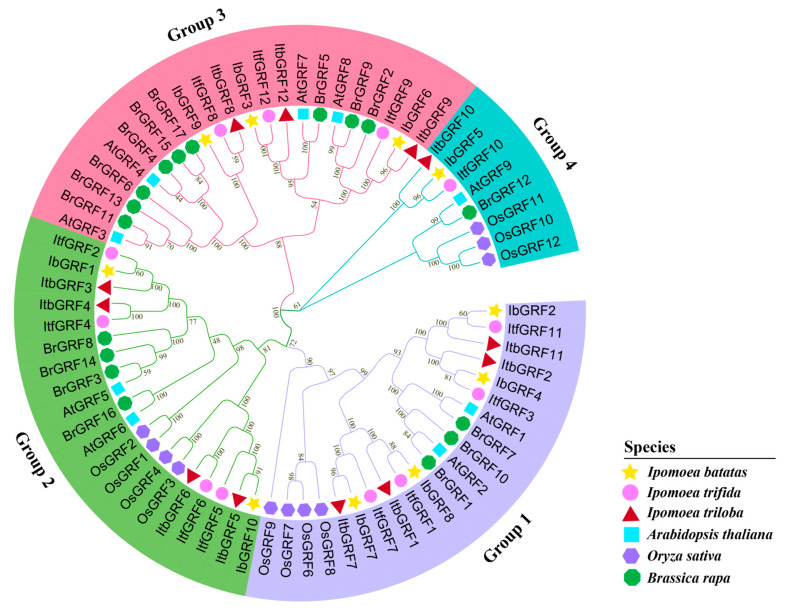
Phylogenetic analysis of *GRFs* in *I. batatas*, *I. trifida*, *I. triloba*, *Arabidopsis*, rice, and Chinese cabbage. A phylogenetic tree was constructed by the neighbor-joining method based on MEGA7.0 with 1000 bootstrap replicates. Different legends represent different species, the yellow stars represent 10 IbGRFs in *I. batatas*, the pink circles represent 12 ItfGRFs in *I. trifida*, the red triangles represent 12 ItbGRFs in *I. triloba*, the blue squares represent 9 AtGRFs in *Arabidopsis*, the purple hexagon represent 10 OsGRFs in rice, and the green octagon represent 17 BrGRFs in Chinese cabbage.

**Figure 4 genes-15-01064-f004:**
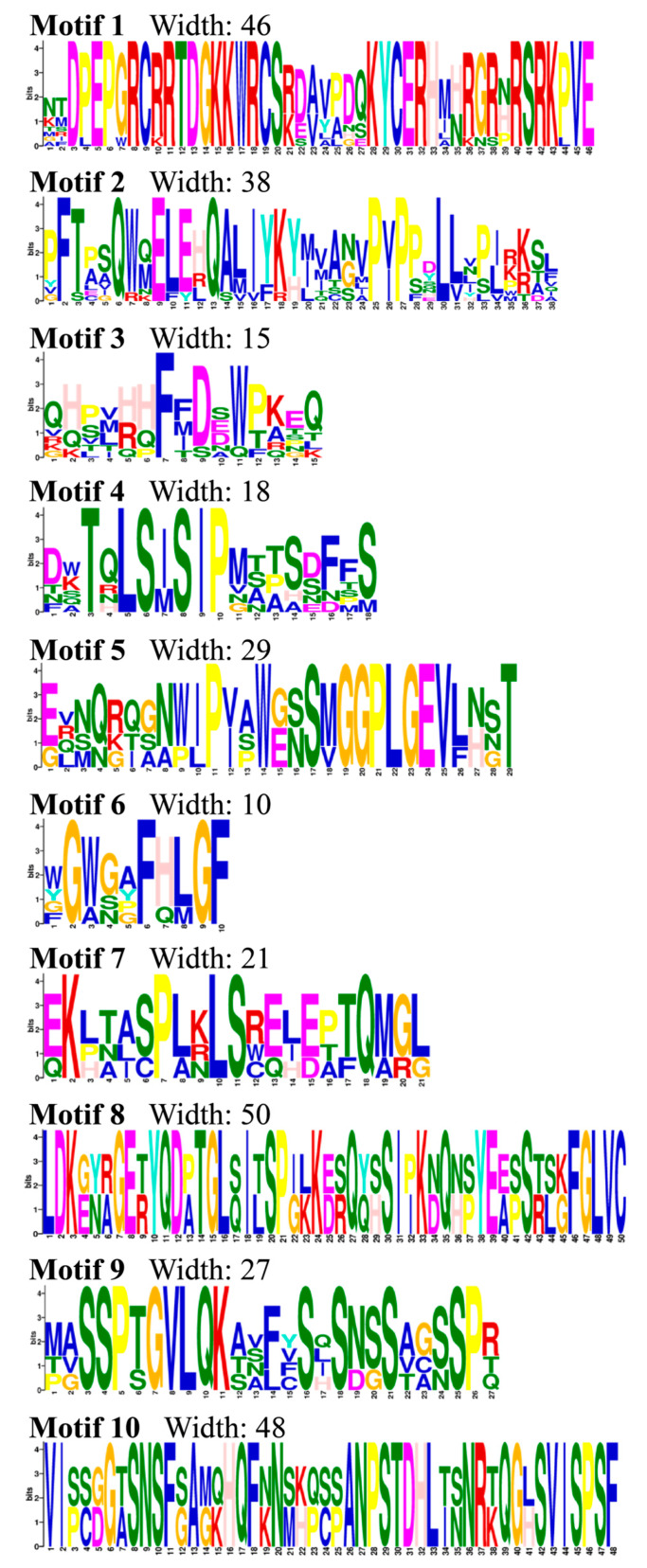
Sequence logos of the 10 conserved motifs.

**Figure 5 genes-15-01064-f005:**
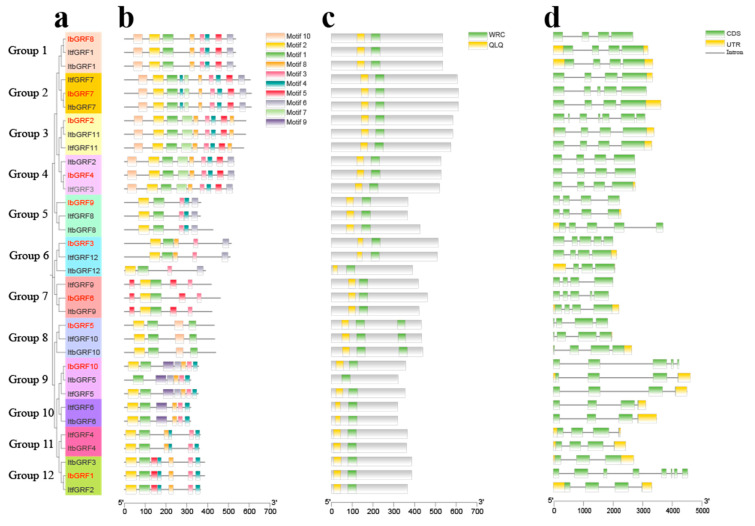
Analysis of conserved motif, conserved domain, and exon–intron structure of *GRF* genes in *I. batatas*, *I. trifida*, and *I. triloba*. (**a**) The phylogenetic tree of *IbGRFs*, *ItfGRFs*, and *ItbGRFs* was constructed by the neighbor-joining method based on MEGA7.0 with 1000 bootstrap replicates. (**b**) The ten conserved motifs were shown in different colors. (**c**) The conserved domains were shown in 34 *GRFs*; the green and yellow boxes represent WRC and QLQ domains, respectively. (**d**) Exon–intron structures of *GRFs*. The green boxes, yellow boxes, and black lines represent UTRs, exons, and introns, respectively.

**Figure 6 genes-15-01064-f006:**
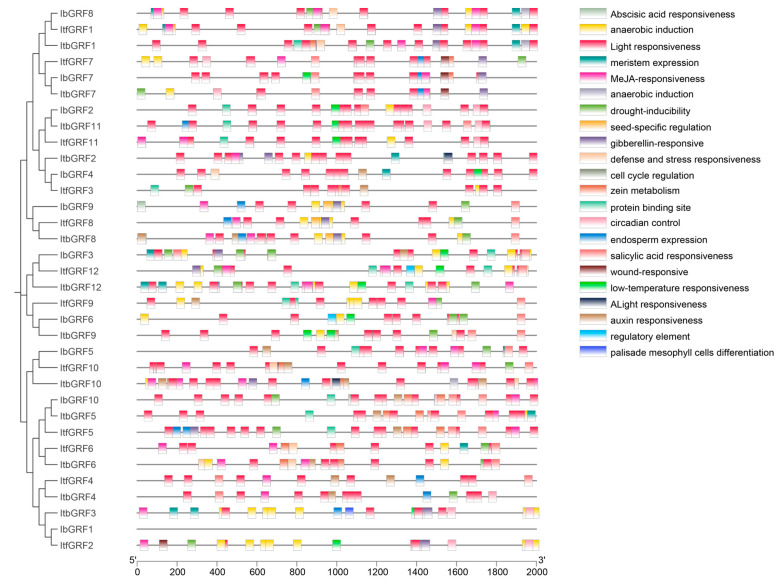
Cis-element analysis in the promoters of *GRFs* in *I. batatas*, *I. trifida*, and *I. triloba*. The cis-elements were divided into twenty-two broad categories. The corresponding color blocks matched to the class of homeopathic components.

**Figure 7 genes-15-01064-f007:**
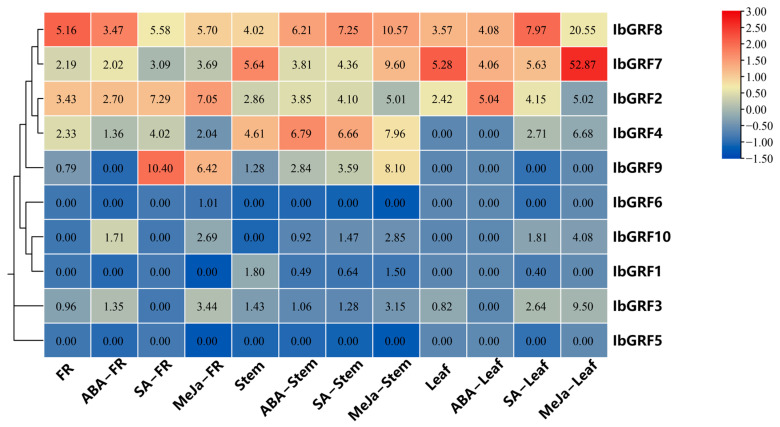
Expression analysis of *IbGRFs* in fibrous roots (FR), stems and leaves of sweet potato under hormones treatment as determined by RNA−seq. The figure value is the true expression, and the scale value is standardized according to the true expression, which reflects the relative expression value.

**Figure 8 genes-15-01064-f008:**
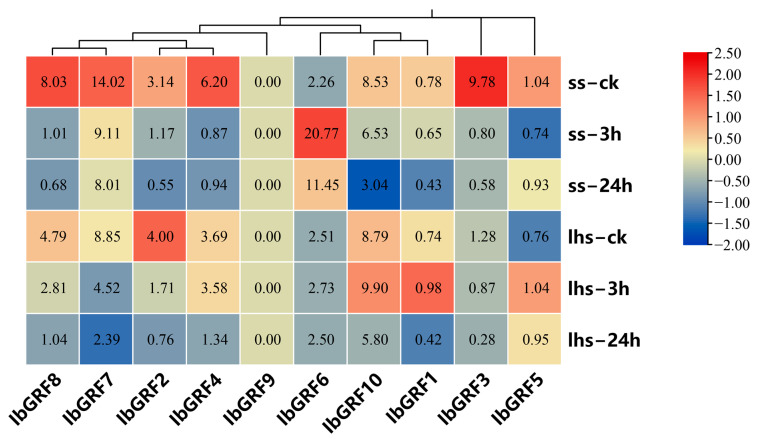
Gene expression patterns of *IbGRFs* under cold stress as determined by RNA−seq. ss: cold−sensitive “Shenshu 28”, lhs: cold−tolerant “Liaohanshu 21”; ck: control group. The figure value is the true expression, and the scale value is standardized according to the true expression, which reflects the relative expression value.

**Figure 9 genes-15-01064-f009:**
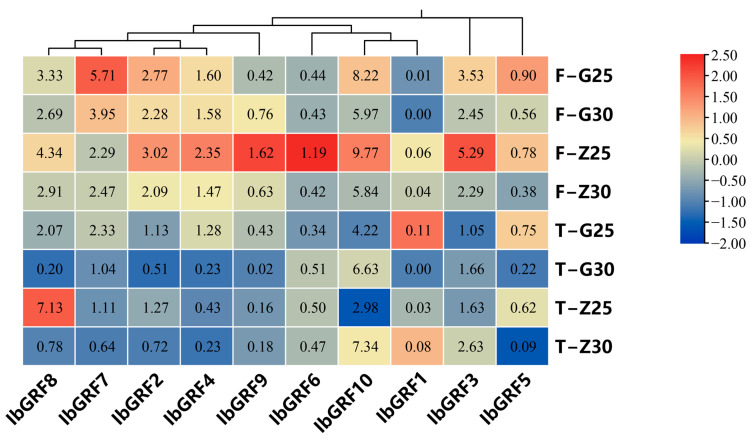
Gene expression patterns of *IbGRFs* under heat stress as determined by RNA−seq. F: fibrous roots; T: tuberous roots; Z: heat−sensitive “Ziluolan”; G: heat−tolerant “Guangshu 87”. The figure value is the true expression, and the scale value is standardized according to the true expression, which reflects the relative expression value.

**Figure 10 genes-15-01064-f010:**
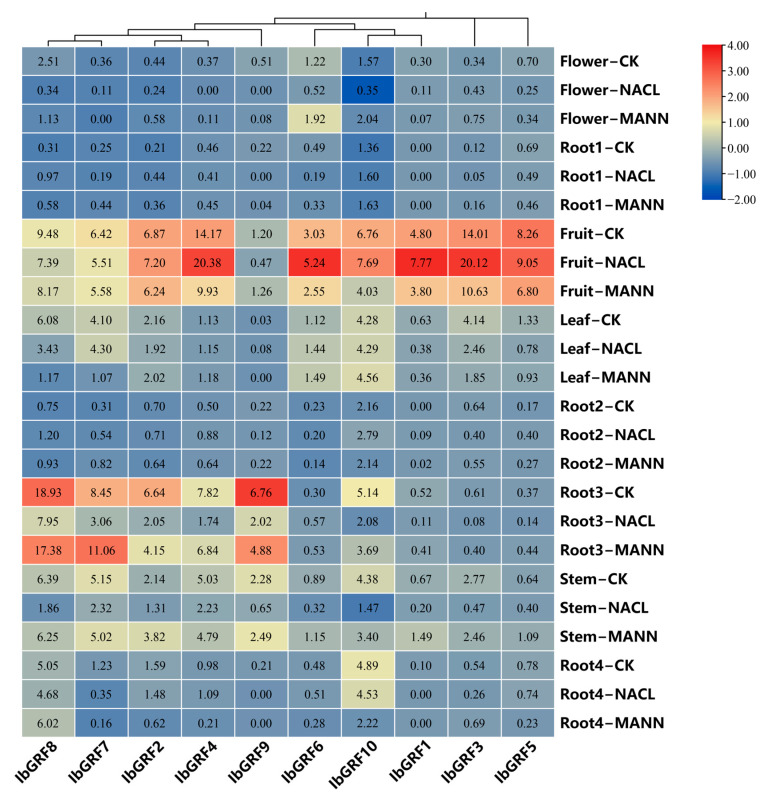
Gene expression patterns of *IbGRFs* in flower, root1 (Firewood root), fruit, leaf, rool2 (fibrous root), rool3 (primary root), stem and rool4 (root tuber) under salt and drought stresses. NACL: salt stress; MANN: drought stress. The figure value is the true expression, and the scale value is standardized according to the true expression, which reflects the relative expression value.

**Figure 11 genes-15-01064-f011:**
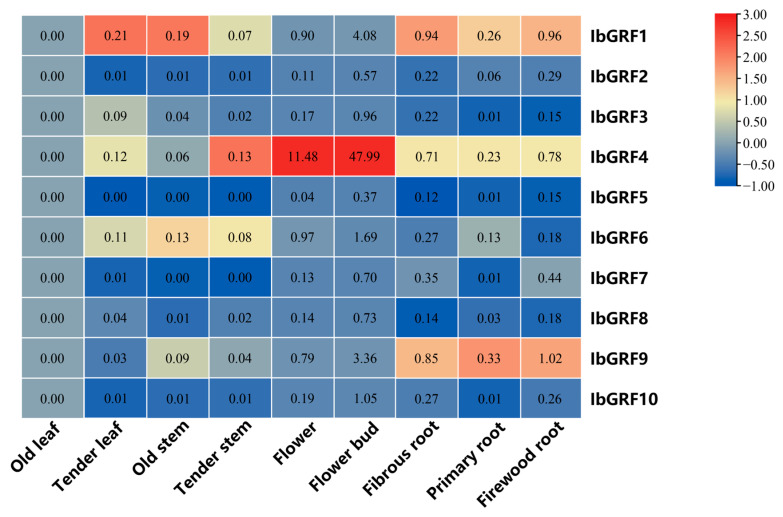
Expression patterns of the 10 *IbGRFs* in 9 different tissues (flower, flower bud, fibrous root, firewood root, primary root, tender stem, old stem, tender leaf, and old leaf). The values were determined via qRT−PCR from three biological replicates consisting of pools of three plants, and the results were subjected to analysis utilizing the comparative CT method. The expression level of *IbGRFs* was analyzed by setting old leaf as control. The fragments per kilobase per million (FPKM) values are shown in the color blocks. The figure value is the true expression, and the scale value is standardized according to the true expression, which reflects the relative expression value.

**Figure 12 genes-15-01064-f012:**
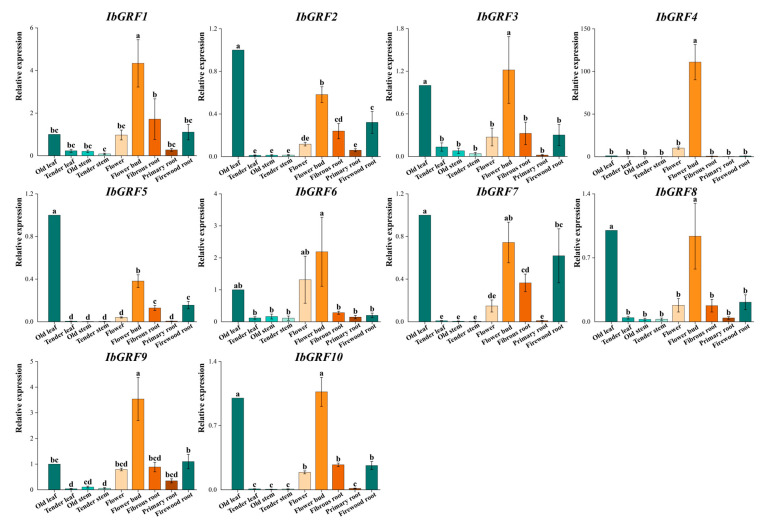
Relative expression levels of 10 *IbGRFs* in 9 different tissues of the sweet potato. The x-axes represent different tissues, including flower, flower bud, fibrous root, firewood root, primary root, tender stem, old stem, tender leaf, and old leaf; the y-axes indicate the relative expression of *IbGRFs*; the error bars depicted represent the standard errors calculated from three technical replicates obtained from a single bulked biological sample. Significant differences in each *IbGRF* at *p* < 0.05 are determined by one-way ANOVA tests, which are indicated with small letters a–e.

**Figure 13 genes-15-01064-f013:**
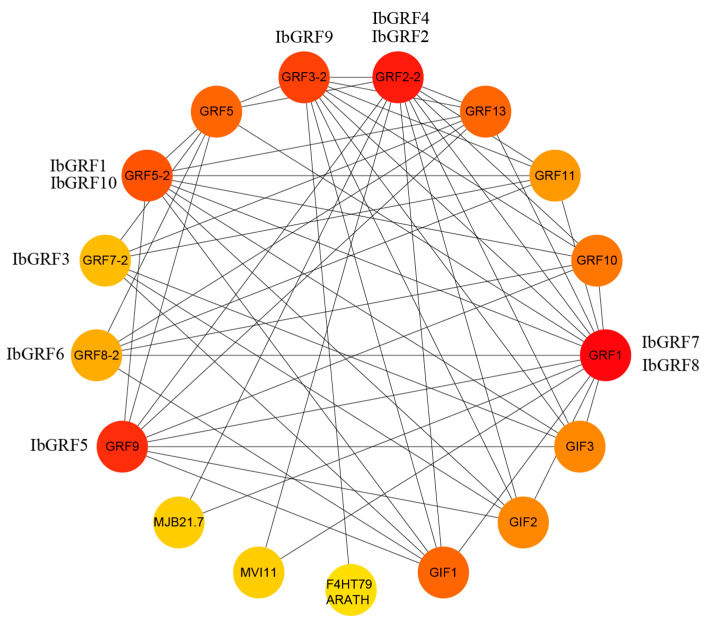
Protein–protein interaction (PPI) network of sweet potato GRF proteins. The depth of the color indicates the degree of interaction between the proteins (the darker the color indicates, the greater the interaction).

**Figure 14 genes-15-01064-f014:**
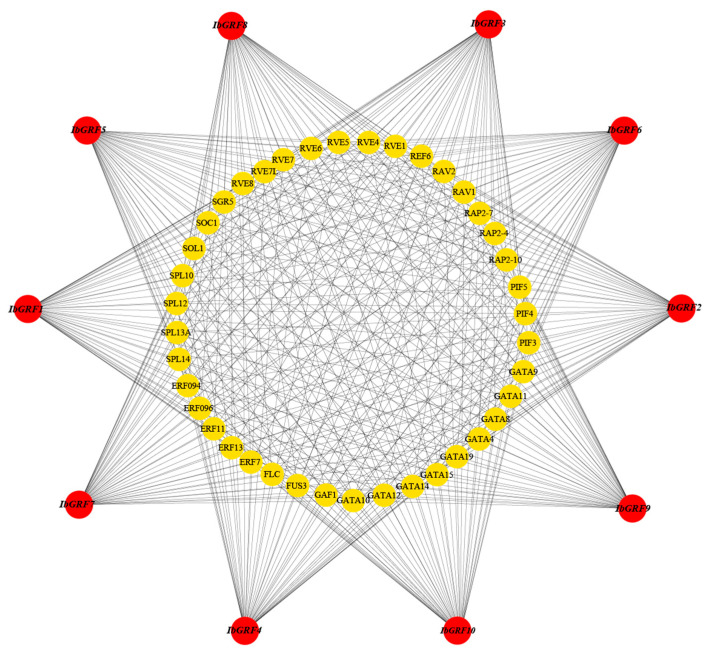
The putative transcription factor regulatory network analysis of sweet potato *GRF* genes. The red circular nodes represent *IbGRFs*, and the yellow nodes in the figure are the transcription factors that interact most with *IbGRFs*.

**Figure 15 genes-15-01064-f015:**
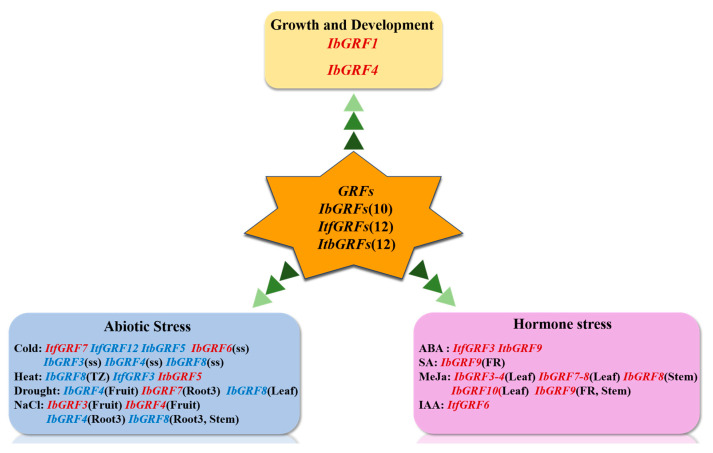
A conclusive graph of functions of *GRFs* in sweet potato and its two relatives, *I. trifida* and *I. triloba*. The red font means the expression is upregulated, and the blue font means the expression is downregulated. The tissue where the gene responds to stress is shown in parentheses. ss: cold-sensitive “Shenshu 28”, TZ: tuberous roots in heat-sensitive “Ziluolan”, Rool3 (primary root), FR: fibrous root.

**Table 1 genes-15-01064-t001:** Identification of *IbGRFs* and analysis of physicochemical properties of proteins in sweet potato.

Sequence ID	Genomic Length (bp)	CDS Length (bp)	Protein Size (aa)	Molecular Weight (kDa)	Isoelectric Point (pI)	Instability Index	Aliphatic Index	Hydropathicity	Subcellular Localization
*IbGRF1*	6470	1158	385	42.99	8.2	59.18	50.99	−0.731	nucl
*IbGRF2*	3970	1752	583	62.20	7.01	46.43	63.09	−0.555	nucl
*IbGRF3*	2902	1542	513	54.62	5.89	60.57	55.59	−0.567	nucl
*IbGRF4*	3050	1584	527	56.49	9.11	58.57	56.51	−0.559	nucl
*IbGRF5*	2444	1296	431	48.02	9.22	53.28	54.94	−0.762	nucl
*IbGRF6*	4991	1383	460	49.72	8.77	54.66	65.78	−0.491	nucl
*IbGRF7*	4380	3672	611	64.87	8.47	49.88	62.52	−0.473	nucl
*IbGRF8*	7869	1605	534	57.54	8.57	45.99	61.63	−0.556	nucl
*IbGRF9*	2473	1104	367	39.79	8.4	55.87	59.13	−0.633	nucl
*IbGRF10*	5407	1071	356	39.16	8.43	57.69	50.67	−0.697	nucl
*ItfGRF1*	3184	1605	534	57.51	8.72	45.53	60.54	−0.578	nucl
*ItfGRF2*	3321	1098	365	40.78	8.52	58.11	49.51	−0.767	nucl
*ItfGRF3*	2756	1560	519	55.58	8.78	58.89	56.99	−0.551	nucl
*ItfGRF4*	2266	1095	364	40.57	7.29	54.87	51.46	−0.887	nucl
*ItfGRF5*	4507	1065	354	38.95	8.43	57.24	50.96	−0.678	nucl
*ItfGRF6*	3116	957	318	35.42	8.39	57.17	48.46	−0.772	nucl
*ItfGRF7*	3339	1815	604	64.02	8.47	49.99	62.12	−0.466	nucl
*ItfGRF8*	2279	1098	365	39.53	7.85	54.56	57.56	−0.653	nucl
*ItfGRF9*	2009	1254	417	45.05	8.87	54.31	55.73	−0.653	nucl
*ItfGRF10*	1979	1302	433	48.28	9.1	55.21	54.92	−0.748	nucl
*ItfGRF11*	3318	1722	573	61.40	6.99	43.62	64.35	−0.541	nucl
*ItfGRF12*	2138	1527	508	54.42	5.89	62.52	55.55	−0.6	nucl
*ItbGRF1*	3355	1605	534	57.61	8.72	45.88	61.44	−0.556	nucl
*ItbGRF2*	2741	1578	525	56.50	8.98	58.43	57.09	−0.565	nucl
*ItbGRF3*	2707	1158	385	42.89	8.46	57.02	52.23	−0.722	nucl
*ItbGRF4*	2440	1083	360	40.26	7.29	53.61	52.03	−0.891	nucl
*ItbGRF5*	4623	960	319	35.24	8.65	59.88	51.66	−0.706	nucl
*ItbGRF6*	3487	957	318	35.36	8.4	55.98	49.69	−0.749	nucl
*ItbGRF7*	3626	1830	609	64.63	8.47	49.76	62.09	−0.469	nucl
*ItbGRF8*	3701	1278	425	46.31	8.24	58.5	59.55	−0.682	nucl
*ItbGRF9*	2213	1263	420	45.38	8.83	54.27	55.31	−0.663	nucl
*ItbGRF10*	2640	1317	438	48.68	9.22	49.34	59.38	−0.658	nucl
*ItbGRF11*	3394	1749	582	62.17	6.79	45.9	62.85	−0.551	nucl
*ItbGRF12*	2075	1170	389	42.70	6.26	67.38	59.2	−0.656	nucl

## Data Availability

Data are contained within the article or Appendix A.

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
