# Peer review of "Genome-Wide Identification and Expression Analysis of Growth-Regulating Factor Family in Sweet Potato and Its Two Relatives"

_genes, 2024, doi:10.3390/genes15081064_

Round 1
Reviewer 1 Report
Comments and Suggestions for Authors
The article is suitable as a scientific idea and implementation, but the final product presented needs a lot of refinement and clarification!
Line 60-61: AtGRF7 Involved in the late expression of the panicles[19]. The statement needs clarification!
Table 1 is not cited in the text!
Line 121: With the molecular weight (MW) was 35.42 kDa to 64.02 kDa, and the isoelectric point (pI) was 5.89(ItfGRF12) to 9.1….The statement needs clarification!
Figure 1. Chromogramal localization and distribution of I.batatas(a), I.trifida(b) and I.triloba(c). The bars represent chromosomes, with chromosome numbers on the left and gene names on the right. There is no word chromogramal!
Line 187: According to the relationship between evolution and development, 70 GRF genes were divided into 4 groups (Groups 1-5 filled with purple, green, pink, and blue, respectively). This sentence under Figure 3 is not in the proper place. This must be in the text, not under the figure!
Figure 4a: The phylogenetic tree of IbGRFs, ItfGRFs, and ItbGRFs. Based on what?
Line 236: As shown in the figure….. Which one Figure?
Lines 240-244: Besides, several abiotic stress-responsive elements, such as anaerobic induction elements (ARE and GC-motif), low-temperature responsiveness element (LIR), drought-inducibility element(MBS), defense and stress responsiveness element(TC-rich repeats) were be found in promoters. The authors write “were be”?!
After Figure 5 appears, Figure 1??? What is written on the abscissa of this figure? The text here is in italics; why?
Line 261-264: Too big sentence and challenging to read! The same goes for the following sentence!
Figure 7 shows a discrepancy between the scale and figure values!
In Figure 8 – What mean ss-ck and ihs-ck?
Figure 12: What is shown on the ordinate? One-way ANOVA with p-value? The standard deviation must be described in the figure.
Figure 13: "The shade of the color represents the degree of interaction."
It is difficult to understand the degree of interaction if there is no legend about what means the different colors!
(Error! Reference source not found.) This appears in many places!
Line 406-407: "The results of TF analysis in IbGRFs showed that a total of 578 TFs were identified and distributed in 45 TF families, and the figure showed only 40..." Which figure?
The discussion is concise! It would be better to expand it further.
Line 543: "Sweet potato plants were cultivated in a field at the experimental field.." better: Sweet potato plants were cultivated at the experimental field..
Reviewer 2 Report
Comments and Suggestions for Authors
The authors have represented the results in a very concise manner.
There are few clarification that needs to be addressed
For Phylogenetic analysis, the tree in Figure 3, does not have bootstrap values. It is very important to add the bootstrap values in the figure, as without the strong bootstrap values, there is no ground support to author's grouping for GRF genes.
In Materials and Methods, in line 471, Please add percentage identity of Blastp to identify GRF proteins.
Authors have used TBtools in various analysis in this manuscript. It would be very helpful for the readers to follow the analysis, if authors have mentioned what parameters or settings they used for all the mentioned analysis.
Comments on the Quality of English Language
A few minor edits is reuired.
Reviewer 3 Report
Comments and Suggestions for Authors
In the manuscript “Genome-Wide Identification and Expression Analysis of Growth-regulating factor Family in Sweet Potato and Its Two Relatives” the authors have conducted a comprehensive study on the Growth-regulating factor gene family in sweet potato and its two diploid relatives. GRF genes play a significant role in plant growth and development, and this research seeks to provide molecular characterization of these genes across the three species. The manuscript is well-structured and provides a detailed analysis of the GRF gene family. The differential expression of IbGRFs in various parts of the sweet potato, especially the upregulation in growth buds, is well documented and provides valuable insights for further functional studies. Methods have been chosen correctly and are appropriate for this type of analysis. Obtained results are scientifically sound and interesting for a broad readers audience. The study provides an in-depth analysis of the physicochemical properties, subcellular localization, and expression patterns of the GRF genes. The differential expression of IbGRFs in various parts of the sweet potato, especially the upregulation in growth buds, is well documented and provides valuable insights for further functional studies. It is noteworthy that all GRF genes identified were located in the nucleus, supporting their role as transcription factors. Ethical issues do not raise concern.
Minor remark:
Line 14 – 15 “In this study, ten, twelve and twelve GRF genes were identified from sweet potato (Ipomoea batatas) and its two diploid relatives (Ipomoea trifida) and (Ipomoea triloba)…”. This sentence should be rewritten mayby like: In this study, ten GRF genes were identified in sweet potato (Ipomoea batatas) ...
In many places, Reference sources were not found.
Line 234 “used PlantCARE online tool..” its necessary to indicate web page of used software
Line 233 “Therefore, we extracted 2000 bp promoter sequence…” and line 506 “The upstream promoter 200bp was extracted…”: disagreement in promoter length.
Round 2
Reviewer 1 Report
Comments and Suggestions for Authors
The article in its present form is suitable for publication after the author's corrections and clarifications! Just some small remarks:
Line 174: "Groups 1-5 filled..." - Must be - Groups 1-4...!
Line 247: "I. batatas, I. trifida and I. triloba" Must be in italic!
In the Discussion section; Line 422 - "3.1. Identification and evolution of sweet potato and two related species trifida and triloba" Will be better "Identification and evolution of sweet potato and its related species I. trifida and I. triloba".
Why batata is in italic (Line 429)?
It is not correct to write just trifida and triloba (Line 427 and others)!
